# Constraint Release Rouse Mechanisms in Bidisperse Linear Polymers: Investigation of the Release Time of a Short-Long Entanglement

**DOI:** 10.3390/polym15061569

**Published:** 2023-03-21

**Authors:** Céline Hannecart, Christian Clasen, Evelyne van Ruymbeke

**Affiliations:** 1Bio- and Soft Matter, Institute of Condensed Matter and Nanosciences, Université catholique de Louvain, 1348 Louvain-la-Neuve, Belgium; 2Soft Matter, Rheology and Technology, Department of Chemical Engineering, KU Leuven, 3001 Leuven, Belgium

**Keywords:** rheology, binary blends of linear polymers, constraint release rouse process

## Abstract

Despite a wide set of experimental data and a large number of studies, the quantitative description of the relaxation mechanisms involved in the disorientation process of bidisperse blends is still under discussion. In particular, while it has been shown that the relaxation of self-unentangled long chains diluted in a short chain matrix is well approximated by a Constraint Release Rouse (CRR) mechanism, there is no consensus on the value of the average release time of their entanglements, $\tau_{obs}$, which fixes the timescale of the CRR relaxation. Therefore, the first objective of the present work is to discuss the different approaches proposed to determine this time and compare them to a large set of experimental viscoelastic data, either newly measured (poly(methyl-)methacrylate and 1,4-polybutadiene blends) or coming from the literature (polystyrene and polyisoprene blends). Based on this large set of data, it is found that with respect to the molar mass of the short chain matrix, $\tau_{obs}$ follows a power law with an exponent close to 2.5, rather than 3 as previously proposed. While this slight change in the power law exponent does not strongly affect the values of the constraint release times, the results obtained suggest the universality of the CRR process. Finally, we propose a new description of $\tau_{obs}$, which is implemented in a tube-based model. The accurate description of the experimental data obtained provides a good starting point to extend this approach to self-entangled binary blends.

## 1. Introduction

The processes involved in the relaxation of the orientation of a monodisperse linear polymer are well identified and understood, based, for example, on the molecular tube picture proposed by Doi, Edwards and de Gennes [1,2]. However, the relaxation process of a long linear polymer moving in a shorter linear matrix is still under discussion [3,4]. Several mechanisms have been proposed to describe the relaxation of the long chains, among which the periodical loosening and reformation of the long chain entanglements involving a short chain (called short-long entanglements), which allows the long chains to further explore their surroundings and relax faster than in the monodisperse case [5,6,7,8]. This relaxation mechanism takes place all along the probe chains and is called Constraint Release (CR). It has been modelled by different approaches, such as the “self-consistent CR” [9], the “dynamic Tube dilation” [10], or the “Double reptation” models [11,12]. For bidisperse polymers containing long chains that are not or barely entangled with other long chains (i.e., long chains are self-unentangled), all the entanglements along the probe chains can be considered as short-long entanglements and the entanglement segments have a similar relaxation time,$\tau_{obs}$, which mostly depends on the relaxation time of the short chains. In cases where the short chain matrix is much shorter than the long probe chains, it has been shown by Graessley [5,13] that it is faster for the long chains to fully relax through these constraint release events, rather than by reptation along their contour length. Consequently, it is assumed that the self-unentangled long chains are fully relaxing via a Rouse process, called the constraint release Rouse (CRR) process [3]. The corresponding terminal relaxation time of the long chains is their CRR time $\tau_{CRR,L}$, which depends on the average waiting time $\tau_{obs}$ for a local CR-jump [3,4,5,6,7,14] (taking place on a distance equal to an entanglement segment), as well as on the number of entanglements along the probe chain *Z_L_* and is defined as:(1)$$\tau_{CRR,L}=\tau_{obs}Z_{L}^{2}$$
$\tau_{CRR,L}$ is usually larger and never shorter than the intrinsic Rouse time of the long chains, $\tau_{R,L}=\tau_{e}Z_{L}^{2}$, where $\tau_{e}$ is the intrinsic Rouse time of an entanglement segment.

It must be noted, however, that the CR process of the long chains contains some non-Rouse features, as discussed in Refs. [3,14] by Watanabe and co-workers. In particular, the eigenfunctions of the chain motion are not following the sinusoidal functions expected for the CRR process. Nevertheless, following Refs. [4,14,15,16,17], the Rouse dynamics can be used, in a first approximation, to describe the relaxation of the long chains diluted in a short chain matrix, as Equation (1) well fits the experimental data, and as the storage and loss moduli of the long component show a Rouse-like relaxation characterized by a power law of around 0.5 when plotted in respect to the angular frequency.

While Equation (1) is well accepted to describe the relaxation time of the long chains, it raises three specific questions, which did not lead to a real consensus up to now and therefore require further investigation: (1) What is the exact criterion to determine if a long chain is slow enough compared to the short linear matrix to relax by CRR, (2) How to determine the value of $\tau_{obs}$ and its dependence on the relaxation time of the short chain matrix, and (3) Is this relationship depending on the chemistry of the bidisperse blends?

In order to address the first question and determine if the molar masses of the short and long components are separated enough to observe the CRR relaxation of the probe chains, Struglinski and Graessley proposed a criterion according to which the renewal of the tube due to the loss/renewal of topological constraints must be faster than the reptation of the long chains in their initial tube [13]:(2)$$r_{SG}=\frac{\tau_{rept,L}}{\tau_{CRR,L}}=\frac{3{\tau_{e}Z}_{L}^{3}}{{\left(3{\tau_{e}Z}_{S}^{3}\right)Z}_{L}^{2}}=\frac{Z_{L}}{Z_{S}^{3}}>1$$
where *Z_S_* is the number of entanglement segments of the short chains, and the ratio $r_{SG}$ is the Struglinski–Graessley parameter. In this definition, the release time of a short-long entanglement segment is assumed to be equal to the reptation time of the short chains: $\tau_{obs}=\tau_{rept,S}=3{\tau_{e}Z}_{S}^{3}$. While this criterion is still often used today, it has been shown both by diffusion experiments [18] and experimental data [3,14,19] that the critical value of $r_{SG}$ at which the CRR relaxation process takes place is, in reality, much shorter than 1. Indeed, diffusion experiments conducted by Green et al. [18] on long dilute deuterated polystyrene (with a molecular weight $M_{L}$) diffusing in polystyrene matrices of various molecular weight $M_{S}$ confirmed the scaling of $r_{SG}\propto \frac{M_{L}}{M_{S}^{3}}$ but showed that the critical $r_{SG}$ value for which the matrix has an impact on the probe chain relaxation is not 1 but $\frac{1}{\alpha_{CR}}$, with $\alpha_{CR}$ being the number of local constraints *per* $M_{e}$ unit related to the efficiency of the CR process. Depending on the definition of the average molar mass between two entanglements $M_{e}$, $\frac{1}{\alpha_{CR}}\approx 0.1$ (if we consider that the plateau modulus $G_{e}=\frac{\rho RT}{M_{e}}$) or 0.064 (if $G_{N}^{0}=\frac{\rho RT}{M_{e}}=\frac{4}{5}G_{e}$). This new criterion has been further tested by Park and Larson [19] within a reptation model on a representative set of polystyrene (PS), polyisoprene (PI), and 1,4-polybutadiene (PBD) bidisperse blends. They found that the critical value $r_{SG}=0.1$ could qualitatively predict whether the long probe chain would reptate in an undilated tube ($r_{SG}<0.1$) or in a dilated tube ($r_{SG}>0.1$) for samples with $r_{SG}$ within a factor 3 away from 0.1, irrespective of the polymer chemistry, and therefore suggesting a universality of this critical value. At the same time, Watanabe et al. conducted an extensive study on PS [6] and PI [14] bidisperse blends and considered the experimentally observed release time of a short-long entanglement segment instead of the bare reptation time of the polymer matrix in the calculation of $r_{SG}$. Based on this wide dataset, they concluded that the critical $r_{SG}$ value for PS blends is $r_{SG}\approx 0.5$, while its value for PI blends is, rather, $r_{SG}\approx 0.2$. This result questions the universality of a critical $r_{SG}$ value and suggests that entanglement dynamics are chemistry-dependent.

More recently, Read et al. [4] proposed to account for the influence of contour length fluctuations (CLF) in the relaxation time $\tau_{d,S}$ of the short chains, in order to establish a more precise expression of the Struglinski–Graessley parameter and defined $r_{SG}^{*}$ as:(3)$$r_{SG}^{*}=\frac{\tau_{rept,L}}{\tau_{d,S}Z_{L}^{2}}=\frac{3\tau_{e}Z_{L}^{3}}{3\tau_{e}Z_{S}^{3}f\left(Z_{S}\right)*Z_{L}^{2}}=\frac{Z_{L}}{Z_{S}^{3}f\left(Z_{S}\right)}\;\mathrm{with}\;f\left(Z\right)=1-\frac{3.38}{\sqrt{Z}}+\frac{4.17}{Z}-\frac{1.55}{Z^{1.5}}$$

In this expression, $f\left(Z\right)$ represents the tube fraction relaxed by CLF at the time the chains relax by reptation [20]. Based on slip-spring simulation results, the authors found the critical value of this new parameter to be $r_{SG}^{*}=0.0254$, i.e., much lower than the initial value of 0.1 established from diffusion experiments for $r_{SG}$. Nevertheless, this new expression contradicts the scaling of $r_{SG}\propto \frac{M_{L}}{M_{S}^{3}}$ found in the other studies. It seems therefore important to further investigate the validity of this criterion, based on a wide set of either new or existing experimental data.

The relationship between $\tau_{obs}$ and $Z_{S}$ is intimately related to the critical value of $r_{SG}$ or $r_{SG}^{*}$. As well illustrated in Ref. [3], based on experimental data, the waiting time for a local hop over a distance of an entanglement segment is not directly proportional to the relaxation time of the short linear matrix. While $\tau_{d,S}\propto M_{S}^{3.5}$ when accounting for the impact of contour-length fluctuations on reptation, diffusion experiments from Green et al. [18] and results of viscoelastic relaxation experiments on PS and PI bidisperse blends from Watanabe et al. [3] suggest that $\tau_{obs}\propto M_{S}^{\alpha }$ with α≈3 and much shorter than the disentanglement time of the short chain matrix. It must be noted, however, that the exact value of the exponent α is not known. For example, in Ref. [14], Sawaka et al. showed that the data available for PI blends do not allow for discrimination between α≈3 and α≈2.8, while it was concluded that α≈2.3, as proposed in Ref. [21], could not be used to accurately describe the data.

To explain this scaling, several physical pictures have been proposed, the first of which is based on the blob theory. Accounting for the number of effective constraints on each entanglement blob, Klein showed that many matrix chains penetrate an entanglement segment, each of them being capable of activating constraint release events [22]. Therefore, the number of constraint release events is enhanced by a factor $1/\sqrt{M_{S}}$, leading to $\tau_{obs}\propto \frac{\tau_{d,S}}{\sqrt{M_{S}}}\propto M_{S}^{3}$. More recently, Shivokhin et al. [15] and Read et al. [4] calculated $\tau_{obs}$ from slip-link simulations and found an empirical expression to determine its value as a function of $\tau_{d,S}$:(4)$$\tau_{obs}=0.047\tau_{d,S}\left(1+\frac{1}{0.36}\sqrt{\frac{\tau_{e}}{0.047\tau_{d,S}}}\right)+\tau_{e}$$

The authors first obtained $\tau_{obs}=0.047\tau_{d,S}$ from Equation (4), but corrected this expression by accounting for the influence of chain friction on the constraint release hop distance: in case of a fast constraint release event, the long probe chain cannot move a significant distance, therefore decreasing the hop length. Consequently, in the case of an intermediate matrix chain length, Equation (4) leads to $\tau_{obs}\propto M_{S}^{3}$ (rather than $\tau_{obs}\propto M_{S}^{3.5}$ for very long matrix chains and $\tau_{obs}\propto M_{S}^{1}$ for short matrix), as expected from experimental data. This expression is interesting, as it suggests that the exponent α can vary depending on the range of molar masses considered for the short chain matrix. However, while this predicts satisfactorily the value of $\tau_{obs}$ for self-unentangled PI bidisperse samples, it remains difficult to explain Equation (4) in the framework of the tube model.

On the other hand, Ebrahimi et al. [16], Lentzakis et al. [23], and Yan et al. [17] proposed a simpler scaling based on the blob picture, experimentally validated for PI, PBD, and poly(hydroxybutyrate) (PHB) star/linear blends, as well as for H/linear and comb/linear blends:(5)$$\tau_{obs}=\frac{3\tau_{d,S}}{Z_{S}^{2}}$$

The authors justified Equation (5) by noticing that when a short matrix chain is relaxed at the time $\tau_{d,S}$, all of its $Z_{S}$ entanglements are relaxed. Therefore, releasing the constraint imposed by a single entanglement only requires the local motion of the chain at the scale of an entanglement blob. However, as mentioned in Ref. [23], it is not clear if this relationship stays valid in the case of a well entangled short chains matrix ($Z_{S}>13$), as it has not been tested in this regime where much larger differences between Equations (4) and (5) appear. Thus, from the above, it is clear that the relationship between $\tau_{obs}$ and $\tau_{d,S}$ needs to be further investigated, using a larger range of matrix molar masses.

Finally, Watanabe and co-workers have shown that, assuming $\tau_{obs}=KM_{S}^{3}$, different values of the proportionality factor K had to be used, depending on the polymer chemistry, in order to describe the experimental data. In particular, it was found that the K constant for PI bidisperse blends is twice as small as the K constant for PS blends [3,14]. This result suggests that the universal behavior of the polymer melts is lost, as their normalized viscoelastic properties cannot be expressed as only a function of their number of entanglement segments and material parameters. It seems therefore important to further investigate this result for polymer blends of other chemistries.

In order to address these questions, the first objective of the present work is to further test and discuss the different scaling that has been proposed to determine $\tau_{obs}$ and its dependency on the polymer chemistry, based on new poly(methyl methacrylate) (PMMA) and 1,4-polybutadiene (PBD) bidisperse blends with self-unentangled (or very poorly entangled) long chains, in order to complement the data of PS [24,25,26,27] and PI [14,28,29,30] blends available in the literature. Based on the CRR time of the long chains, as well as $\tau_{obs}$ determined from these data, we would like to discuss the relationship between $\tau_{obs}$ and the number of entanglements of the short chain matrix $Z_{S}$ and the value of the Struglinski–Graessley parameter [3,4,5,13,31].

Our second objective is to propose a simple expression to determine $\tau_{obs}$ that can easily be incorporated in a CRR model and to validate it by comparing the theoretical results to the viscoelastic data of dilute binary blends of various chemistries.

The manuscript is organized as follows: in Section 2, a model is proposed to account for the CRR process underwent by a long chain well diluted in a short linear matrix. Section 3 presents the PMMA and PBD samples measured in this work, as well as all the other blends found in the literature, with self-unentangled long chains. In Section 4, the values of $\tau_{obs}$ are first extracted from linear viscoelastic data of those samples and then discussed in relation to the molar mass and relaxation time of the short chain matrix. The influence of the polymer chemistry, i.e., of PS, PI, PMMA, and PBD samples, is also investigated. Based on these results, a critical value for the Struglinski–Graessley criterion is proposed. Then, in Section 5, we include the CRR process in a tube model and validate it with the experimental data. Finally, conclusions are presented in Section 6.

## 2. Modeling

### 2.1. Description of the CRR Model

In this section, we focus on modeling the linear viscoelastic behavior of bidisperse linear blends in which the long chains are self-unentangled and the molar masses of the long and short components are well separated, such that the Struglinski–Graessley criterion is fulfilled. Under these conditions, one can assume that the long chains only relax by a CRR process governed by the time of the release/formation of the short-long entanglements, $\tau_{obs}$ [6,7,8]. Thus, as illustrated in Figure 1, two Rouse processes are observed in the relaxation of the long chains. First, at very short times, the probe chain relaxes its orientation by successive (intrinsic) Rouse relaxation modes *p* involving longer and longer molecular segments. However, at time *t* = $\tau_{e}$, at which the chain is relaxed at the length scale of the entanglement segments of mass $M_{e}$ (see Figure 1a), intrinsic Rouse relaxation cannot take place anymore, due to the entanglement constraints imposed on the chain. Since $\tau_{e}$ corresponds to the relaxation time of the mode ${p=Z}_{L}$ (the number of entanglements along the long probe chain), the first component $G_{R,L}(t)$ of the viscoelastic relaxation modulus of the probe chain corresponding to this intrinsic Rouse relaxation process can be expressed as the sum of the contribution of each Rouse mode [20]:(6)$$G_{R,L}(t)=\frac{\upsilon_{L}\rho RT}{M_{L}}\sum_{p=Z_{L}+1}^{N}e^{-2tp^{2}/\tau_{R,L}}$$
where $\upsilon_{L}$ is the weight fraction of the long chains, ρ is the polymer density, T the temperature, R the universal gas constant, and *M_L_* the molar mass of the polymer.

As mentioned in the Introduction, the relaxation of molecular segments longer than the entanglement segments can only take place at times longer than the release time of the short-long entanglements, $\tau_{obs}$ (with $\tau_{obs}\geq \tau_{e}$). Thus, no relaxation takes place between $\tau_{e}$ and $\tau_{obs}$, and the CRR time of the whole chain, which evolves at the rhythm of the short-long entanglements disentanglement/re-entanglement, is equal to ${\tau_{CRR,L}=\tau }_{obs}Z_{L}^{2}$, rather than its intrinsic Rouse time, $\tau_{R,L}=\tau_{e}Z_{L}^{2}$. It must be noted, however, that long chains diluted in an oligomeric matrix (such that $\tau_{d,S}<\tau_{e}$) fully relax by their intrinsic Rouse process (thus $\tau_{obs}=\tau_{e}$ and $\tau_{CRR,L}=\tau_{R,L}$). Accounting for this condition, the CRR process of self-unentangled long chains can be approximated as [32]:(7)$$G_{CRR,L}(t)=\frac{\upsilon_{L}\rho RT}{M_{L}}\sum_{p=1}^{Z_{L}}e^{-2tp^{2}/\tau_{CRR,L}}$$

Then, combining Equations (6) and (7), the relaxation modulus of the long chains is well described as:(8)$${G_{L}\left(t\right)=G_{R,L}\left(t\right)+G}_{CRR,L}(t)$$

In the case of self-entangled long chains, the CRR process stops once the long-long entanglements (of mass $M_{e}/\upsilon_{L}$) are relaxed, i.e., at time *t* = $\tau_{obs}/\upsilon_{L}^{2}$. This relaxation time corresponds to the CRR mode ${p=\upsilon }_{L}Z_{L}.$ The relaxation of the longer modes $\left(1\leq Z_{L}<\upsilon_{L}Z_{L}\right)$ takes place via other relaxation processes, such as the reptation and contour length fluctuations. In this study, the long probe chains in some bidisperse samples are barely entangled with other long chains ($1<\upsilon_{L}Z_{L}<3$), leading to a very small portion of probe chain actually constrained by long-long entanglements. In the model, we neglect these long-long entanglements and assume that the relaxation of these poorly self-entangled long chains is fully described by a CRR process. As discussed in Section 5, this assumption can lead to a slightly underestimated terminal relaxation time for specific blends.

The relaxation modulus of the blend also includes the contribution from the short chain matrix, $G_{S}(t)$. To determine the latter, we assume that the disorientation processes of the short matrix chains in the blend are unaffected by the presence of the long chains, i.e., the viscoelastic relaxation modulus of the short chain matrix in the blend is the same as in the monodisperse state. Such an assumption is justified, since the concentration of the long component is very small. $G_{S}\left(t\right)$ is calculated as explained in detail in Ref. [33]; if the short chains are unentangled ($M_{S}\leq {2M}_{e}$), the matrix fully relaxes by a Rouse process:(9)$$G_{S}\left(t\right)=\frac{\upsilon_{S}\rho RT}{M_{S}}\sum_{p=1}^{N}e^{-2tp^{2}/\tau_{R,S}}$$
where $\upsilon_{S}$ represents the weight fraction of the short chains. If the short chains are entangled, their relaxation modulus is determined based on the simplified time marching algorithm (TMA) [34,35]:(10)$$G_{S}\left(t\right)=\frac{4}{5}\frac{\upsilon_{S}\rho RT}{M_{e}}\varphi_{S}\left(t\right)\Phi_{TB,eff}(t)$$
where $\varphi_{S}(t)$ is the survival fraction of the initial entanglement segments along a short chain, considering that the chains can relax by reptation and contour length fluctuations, and the function $\Phi_{TB,eff}(t)$ accounts for both the tube dilation factor (when $\Phi_{TB,eff}\left(t\right)\leq 1$) and the intrinsic Rouse relaxation of the entanglement segments ($\Phi_{TB,eff}\left(t\right)\geq 1$ for $t\leq \tau_{e}$) [33]:(11)$$\Phi_{TB,eff}\left(t\right)=max\left(\upsilon_{L}+\upsilon_{S}\varphi_{S}(t),\frac{5}{4}\sqrt{\frac{\tau_{e}}{2t}}\right)$$

The viscoelastic modulus of the whole sample finally results from the sum of the contributions of the long and short chains:(12)$$G_{blend}\left(t\right)=G_{L}\left(t\right)+G_{S}\left(t\right)$$

The complex viscoelastic spectrum $G^{*}\left(\omega \right)$ is then obtained from $G_{blend}\left(t\right)$ using the Schwarzl equations [36].

### 2.2. Validity of the CRR Model

While similar approaches have already been used in previous works and showed good agreement with the data, it is important to note that Equation (6) is derived from the assumption that the CRR process taking place at the local level does not depend on the global motion of the chain, which is most likely not the real case. Furthermore, as detailed in Refs. [3,14,37,38,39], the CR-Rouse feature of the chain motion is only valid if we can consider that the segments between two entanglements always have the same length during the chain relaxation. Indeed, if the number of monomers between two entanglements varies, this directly leads to a tension-equilibration process, i.e., to a transfer of monomers along the contour length, which speeds up the relaxation of the entanglement segments located near to the chain ends. As discussed in Ref. [37], this may be the reason why non-Rouse features are observed in the dielectric data of dilute PI blends. This possible faster relaxation of the chain ends is not accounted for in the present approach. It is, however, expected that this should not significantly affect the predicted curves, as the tension equilibration is a global process that involves the motion of the whole chain and is therefore rather slow in comparison to the local CR-Rouse equilibration of the chain.

Another issue, which has been raised in Ref. [14], is the validity of the CRR model at short times, $t<\tau_{d,S}$. As it is shown in Section 4, the average release time of a short-long entanglement segment, $\tau_{obs}$, is much shorter than the average relaxation time of the short chain matrix. Therefore, at the time at which we consider the relaxation of the faster modes of the CRR process to take place, a fraction of the initial short-long entanglements are still existing, preventing the local equilibration of the long chains. However, at long times, this process is averaged, and the proposed assumption is acceptable. Deviations are thus expected at short times, and, in particular, between $\tau_{obs}$ and $\tau_{d,S}$. This point is further investigated in the Appendix A (see Appendix A).

## 3. Materials and Methods

### 3.1. Bidisperse Blends Composed of Self-Unentangled Long Chains

*PMMA bidisperse blends*: Poly(methyl methacrylate) of various molar mass, with a high syndiotactic ratio (>79%) and low polydispersity index (PDI), were commercially obtained from Polymer Source, Inc. (Montreal, QC, Canada). The weight average molar mass M of the materials has been measured with size-exclusion chromatography (SEC column by Agilent, Santa Clara, CA, USA) and the $T_{g}$ of the monodisperse samples has been determined by differential scanning calorimetry (DSC) with a standard Heat-Cool-Heat procedure (heating rate of 10 K/min under inert atmosphere), in a Q2000 instrument (TA instruments, New Castle, DE, USA). The main characteristics of the blends, as the molecular weight of the long chain $M_{L}$ and its weight fraction $\upsilon_{L}$, as well as of the short chain matrices (their molecular weight $M_{S},$ polydispersity index (PDI), and $T_{g}$), are given in Table 1.

The bidisperse PMMA blends composed of 2 or 3 wt% of PMMA234 were prepared either by precipitation for high molecular weight matrices (PMMA27, PMMA35, and PMMA60) or by dilution for shorter matrices. To this end, the components of the blend were first weighed to obtain the desired weight fraction and dissolved together in tetrahydrofuran (THF, purchased from Merck KGaA, Darmstadt, Germany) to obtain a concentrated polymer solution of 30 mg/mL. The mixture was then stirred slowly overnight at room temperature. The solutions containing the high molecular weight matrices were precipitated drop by drop in a large amount of methanol under continuous stirring. After filtration, the obtained powder was dried in a vacuum oven set at 70 °C for 5 days to remove residual solvent. The other solutions were poured into a form made of thick aluminum foil and covered with a thin perforated aluminum foil. They were then left to dry in a fume hood until a solid film had formed (>24 h). Flakes of this thin film were then placed to dry in the vacuum oven set at 50 °C for 7 days to remove remaining solvent. The weight fraction of each blend was verified by SEC and are listed together with the blend molecular characteristics in Table 1.

*Bidisperse PBD blends*: 1,4-Polybutadiene of varying molar mass and low polydispersity index (PDI), were commercially obtained from Polymer Standards, GmbH (Mainz, Germany).

Dilute bidisperse PBD blends were prepared from these samples, and their molecular characteristics are listed in Table 2. PBD254 was first diluted in THF to a concentration of 2 mg/mL, and an appropriate amount of this solution was dropped in solutions of 200 mg/mL of PBD matrices in THF to obtain the desired blend weight fraction. An antioxidant (butylated hydroxytoluene, purchased from Merck KGaA, Darmstadt, Germany) was added in a small amount to each solution (~0.5 wt%) prior to manual stirring. The PBD blends were then left to dry under increasing vacuum conditions, at room temperature and in the dark, for 9 to 12 days.

*Bidisperse samples from the literature*: A wide set of linear viscoelastic data of linear bidisperse blends of PS and PI (with similar cis-1,4:trans-1,4:3,4 ratio) with not or poorly self-entangled long chains ($\upsilon_{L}Z_{L}<3$) has been studied in the literature. These samples are listed in Table 3, Table 4, Table 5 and Table 6, depending on the sample chemistry. Main characteristics and corresponding references are given. For each sample, when available, the value of the glass transition temperature ($T_{g}$) that has been used in the original reference is given. When needed, the theoretical value of $T_{g}$, determined based on the Fox-Flory Equation (see Section 4.1), is also shown.

Regarding PBD samples from the literature, full viscoelastic relaxation curves were found for only three sets of bidisperse linear blends containing self-unentangled long chains, PBD550-20 [40], PBD208-15, and PBD412-15 [41] (see Table 5). For these last two samples, only the contribution of the long chain to the viscoelastic modulus is reported. On the other hand, an extensive study from Wang et al. [41] on linear PBD bidisperse blends provides the zero-shear viscosity data of many other sets of samples. These samples are also used in Section 4, and their characteristics are listed in Table 6.

### 3.2. Linear Viscoelastic Measurements

Prior to the linear viscoelastic measurement, PMMA samples were dried under vacuum overnight and molten, pressed and annealed under vacuum at $T=T_{g}+70\;{}^{\circ}C$ in disks of 5.5 mm diameter and 1.4 mm height, yielding to 0.6 mm thick 8 mm disks, while the PBD samples were loaded at room temperature on the 8 mm plate to obtain a thickness between 0.5 and 1 mm between both plates. Upon a progressive decrease of temperature, the gap between both plates was manually decreased to ensure full contact between both plates and the sample.

The small amplitude oscillatory shear behavior of these samples was measured on an MCR 301 rheometer (Anton Paar GmbH, Graz, Austria) for PMMA and on an AR-2000 rheometer (TA Instruments, New Castle, DE, USA) under $N_{2}$ atmosphere for PBD. A stainless steel 8 mm parallel plate geometry was used, with a convection oven for temperature control. For each sample, an amplitude sweep was performed before each measurement to determine the linear region and choose the appropriate imposed deformation (between 0.1 and 10%). Frequency sweeps were then performed between 200 and 0.01 rad/s at temperatures ranging from 200 to 120 °C for PMMA samples and between 100 and 0.01 rad/s at temperatures ranging from 40 to −80 °C for PBD samples. To avoid crystallization, PBD samples were regularly brought back to 30 °C for a few minutes before conducting measurements at the lowest temperatures.

## 4. Results and Discussion

### 4.1. Linear Viscoelastic Data

First, the master-curves built for the PMMA samples are shown in Figure 2. Since the samples containing the shortest matrices (PMMA3 and PMMA15) have a lower $T_{g}$, the reference temperature of the PMMA blends has been adjusted to ensure iso-$T_{g}$ conditions, $T_{ref}=T_{g}+60\;{}^{\circ}C$ (see Table 1). Utilizing the appropriate reference temperatures, the shift factors of the different samples follow the same WLF Equation (13), with $c_{1}=7.22$ and $c_{2}=136.5\;{}^{\circ}C$.

Regarding the PBD samples, the frequency-dependent storage and loss moduli obtained at different temperatures were horizontally shifted to a reference temperature $T_{ref}=-50\;{}^{\circ}C$. Results are shown in Figure 3 for both the monodisperse samples and the corresponding blends. The shift factors used to build the master-curves are shown in the insert of the figure. It is observed that all master-curves superimpose well at high frequencies, confirming that the samples all have a similar glass transition temperature. Moreover, the shift factors used for building the different master-curves follow well the William–Landel–Ferry (WLF) equation [42,43]:(13)$$\log_{10}\left(a_{T}\right)=\frac{-c_{1}\left(T_{data}-T_{ref}\right)}{c_{2}+\left(T_{data}-T_{ref}\right)}$$
with $c_{1}=6.66$ and $c_{2}=93.9\;{}^{\circ}C$ at $T_{ref}=-50\;{}^{\circ}C$.

For both sets of samples, it is observed that the influence of the few percent of the long component on the viscoelastic response of the short chain matrix is negligible, as it has been assumed in the model. It is also observed that, after the relaxation of the short chain matrix, the storage modulus of most of the blends decreases with a slope of ½, which well corresponds to a CRR regime, as further discussed in Section 4.3.

In order to analyze and compare the linear viscoelastic data of the blends coming from the literature (see Table 3, Table 4, Table 5 and Table 6), the viscoelastic curves for each different set of samples were shifted at the same distance from their glass transition temperature ($T_{g}$), to ensure that they are all characterized by the same Rouse time of an entanglement segment, $\tau_{e}$ [44]. Since, for the PS samples (see Table 3), the value of $T_{g}$ was not given, we used the Fox-Flory equation to determine their value for the short chain matrices [45]:(14)$$T_{g}\left(M\right)=T_{g,\infty }-\frac{M_{ref}}{M}[{}^{\circ}C]$$
where $T_{g,\infty }=106.6\;{}^{\circ}C$ is the glass transition temperature of an ultra-high molecular weight PS polymer, and $M_{ref}$ = 1.1 × 10^5^ g/mol for PS [45]. Then, the glass transition temperature of the blends was determined as:(15)$$\frac{1}{T_{g,blend}}=\frac{\upsilon_{L}}{T_{g,L}}+\frac{1-\upsilon_{L}}{T_{g,S}}$$
where the longest blend component can be assumed to be long enough to have $T_{g,L}=T_{g,\infty }$.

Knowing the $T_{g}$ value of the blends, the horizontal shift to apply to the data in order to compare them at iso-$T_{g}$ was determined based on the WLF equation. For the PS samples, the constants $c_{1}$ and $c_{2}$ were fixed to $c_{1}=6.74$, $c_{2}=133.6\;{}^{\circ}C$ for PS at ${(T}_{ref}-T_{g})=60.4\;{}^{\circ}C$, following Refs. [24,25]. A similar approach was applied to the PBD samples of Ref. [41], which were all measured at 40 °C (see Table 6). The appropriate $T_{ref}$ was evaluated from the values of $T_{g}$ reported for the monodisperse samples and from Equation (15), and we used $c_{1}=3$ and $c_{2}=180\;{}^{\circ}C$ at ${(T}_{ref}-T_{g})=142\;{}^{\circ}C$ from Ref. [46] to shift the values of the zero-shear viscosity at iso-$T_{g}$ conditions, knowing that $\eta_{0}(T_{ref})=\underset{\omega \to 0}{\lim}\frac{G_{T_{data}}^{\prime \prime }\left(\omega \right)}{\omega_{T_{data}}a_{T}}=\eta_{0}(T_{data})/a_{T}$.

As Equations (13)–(15) are empirical, the validity of the shifting has been checked, based on the rheological curves. For the PS samples, it was found that the complex moduli well superimpose in the high frequency Rouse regime, as illustrated in the Appendix A (see Appendix A) for the monodisperse matrices and some of the blends. It must be noted that for samples PS2810-23.4 and PS2810-39 (see Table 3), a better agreement at high frequency was found by using the theoretical $T_{g}$ values instead of the experimental data. For the PBD blends of Ref. [41], the shifting could not be validated because the storage and loss moduli are not reported. This uncertainty must be taken into account in the analysis of this data.

### 4.2. Material Parameters

In order to analyze and model the viscoelastic data, first one needs to determine the material parameters, i.e., the molar mass of an entanglement segment $M_{e}$, its intrinsic Rouse time $\tau_{e}$, and the plateau modulus $G_{N}^{0}$. These parameters are listed in Table 7. They have been chosen in order to best fit with TMA experimental data of the monodisperse samples investigated in this work. It should be noted that the parameters employed to model the PI samples at 40 °C are the same as those used in Ref. [47], and the parameters used to model the PBD samples at 25 °C are the same as those employed by van Ruymbeke et al. [48].

To confirm the values of the entanglement molecular weight $M_{e}$ chosen in Table 7 for the datasets considered in this article, we follow the method described in Ref. [47] and compare the viscoelastic data normalized by $G_{N}^{0}$ and plotted against $\omega \tau_{e}$ of at least one monodisperse sample from each set of blends [14,24,25,26,27,28,29,30,40,41] to another data set with a different polymer chemistry but that supposedly has the same number of entanglements Z. Assuming that the normalized linear viscoelastic properties only depend on the number of entanglements, $G^{\prime }$ and $G^{\prime \prime }$ should superimpose onto a single curve for samples with the same Z. This is indeed verified, as demonstrated in Figure 4, for Z= 3, 4, 5, 6, 9, 11, 12, and 26.

The good superposition of the normalized data provides relationships between the entanglement molecular weight of different chemistries: $Z=\frac{M\left(PS\right)}{M_{e}^{PS}}=\frac{M\left(PI\right)}{M_{e}^{PI}}=\frac{M\left(PMMA\right)}{M_{e}^{PMMA}}=\frac{M\left(PBD\right)}{M_{e}^{PBD}}$. After averaging the values on the different sets of data, we obtain:(16a)$$M_{e}^{PS}/M_{e}^{PI}\approx 3.96$$
(16b)$$M_{e}^{PS}/M_{e}^{PMMA}\approx 2.00$$
(16c)$$M_{e}^{PS}/M_{e}^{PBD}\approx 8.27$$
(16d)$$M_{e}^{PI}/M_{e}^{PBD}\approx 2.09$$

Assuming that $M_{e}^{PS}=14.00\;\mathrm{kg}/\mathrm{mol}$, these equations lead to $M_{e}^{PI}=3.54\;\mathrm{kg}/mol$, $M_{e}^{PMMA}\approx 7.00\;\mathrm{kg}/mol$ and $M_{e}^{PBD}\approx \;1.69\mathrm{kg}/mol$, which is close to the values proposed in Table 6.

### 4.3. Determination of the CRR Time of the Long Chains, τ_CRR,L_

In order to determine the value of $\tau_{obs}$, we determine $\tau_{CRR,L}$ from the experimental storage and loss moduli of the blends, $G_{blend}^{\prime }\left(\omega \right)$ and $G_{blend}^{\prime \prime }\left(\omega \right)$, following Ref. [3]. This requires first removing the short chain matrix contribution:(17)$$G_{L}^{\prime }\left(\omega \right)=G_{blend}^{\prime }\left(\omega \right)-{\upsilon_{S}G}_{S,mono}^{\prime }\left(\omega \right)$$
(18)$$G_{L}^{\prime \prime }\left(\omega \right)=G_{blend}^{\prime \prime }\left(\omega \right)-{\upsilon_{S}G}_{S,mono}^{\prime \prime }\left(\omega \right)$$
where $G_{S,mono}^{\prime }\left(\omega \right)$ and $G_{S,mono}^{\prime \prime }\left(\omega \right)$ are the experimental storage and loss moduli of the matrix in the monodisperse state. Then, assuming that the long chains relax by a CRR process, the CRR time of the long component is determined from the following low-frequency limit:(19)$$\tau_{d,L}=\underset{\omega \to 0}{\lim}\frac{G_{L}^{\prime }\left(\omega \right)}{\omega G_{L}^{\prime \prime }\left(\omega \right)}=\frac{\int_{0}^{\infty }tG_{L}\left(t\right)dt}{\int_{0}^{\infty }G_{L}\left(t\right)dt}=\frac{\tau_{CRR,L}}{2}\frac{\sum_{p=1}^{Z_{L}}1/p^{4}}{\sum_{p=1}^{Z_{L}}1/p^{2}}\approx \frac{\tau_{CRR,L}}{2}\frac{\pi^{2}}{15}$$

In order to validate the values found for $\tau_{CRR,L}$ with this equation, we compare in Figure 5 the contribution of the long chains to the storage modulus of the blends, $G_{L}^{\prime }$ (see Equation (17)), vertically shifted by a factor $\frac{\upsilon_{L}\rho RT}{M_{L}}$ and horizontally shifted by a factor $\frac{\pi^{2}}{30}\tau_{CRR,L}$. The data corresponding to the blends PI308-94, PI626-179, and PS316(10 wt%)-89 have been removed, as their long component do not relax by a CRR process (see Section 5). Despite some data scattering observed at high frequency (resulting from the removal of the matrix contribution obtained from experimental data), the low frequency data well superimposes for all blends, and a Rouse-like relaxation is observed, immediately followed by the terminal regime of relaxation. Moreover, their terminal regime well follows the theoretical curves corresponding to $\tau_{CRR,L}=1$ (and assuming that $G\left(t\right)=\sum_{p}e^{-2p^{2}t}$; see the black curves). This confirms that the long linear chains are relaxing via a CRR process. The viscoelastic data also confirms the CRR relaxation of the long chain for PBD208-15 and PBD412-15 diluted at 2 or 0.5–1 wt%, respectively, (see Table 5) and measured by Wang et al. [41] at 40 °C, as can be seen in Figure 5a. Indeed, these data superimpose well with the series of PBD254 diluted in various matrices and measured at −50 °C. This suggests that the blends composed of shorter matrices should also relax by a CRR-like motion and can be considered in our analysis of the CRR time.

In the case of the PBD blends presented in Table 6, as only the zero-shear viscosity data are available, [41] the value of $\tau_{CRR,L}$ is determined based the following approximation (after removal of the matrix contribution from the experimental data, and under iso-$T_{g}$ condition):(20)$$\eta_{0,L}=\underset{\omega \to 0}{\lim}\frac{G_{L}^{\prime \prime }\left(\omega \right)}{\omega }=\int_{0}^{\infty }G_{L}\left(t\right)dt=\frac{\upsilon_{L}\rho RT}{M_{L}}\frac{\tau_{CRR,L}}{2}\sum_{p=1}^{Z_{L}}\frac{1}{p^{2}}\approx \frac{\upsilon_{L}\rho RT}{M_{L}}\frac{\tau_{CRR,L}}{2}\frac{\pi^{2}}{6}$$

This method is, however, more approximate than the former one, as it involves the sample density, ρ, which is not accurately known [47] and is dependent on the assumption that the long chain fully relaxes by a CRR motion. However, this assumption cannot be validated because the storage and loss moduli data are not available. Nevertheless, it can be noted that if the long chains relax only by a CRR process, their corresponding relaxation time $\tau_{CRR,L}$ should not depend on the weight fraction of the long chains, $\upsilon_{L}$. This was, indeed, observed (see Figure 6), within a ±10% difference for $\tau_{CRR,L}$, which supports this method. In the case of PBD208-15 and PBD412-15, we determined $\tau_{CRR,L}$ both from the long chain contribution to the viscoelastic relaxation modulus and from the zero-shear viscosity. Comparison between the values obtained with the two methods led to similar results (within 20% uncertainty).

### 4.4. Relationship between τ_obs_ and Z_S_

The release time of a short-long entanglement segment, $\tau_{obs}$, is determined from $\tau_{CRR,L}$ and $Z_{L}$ (see Equation (1)), with $Z_{L}=M_{L}/M_{e}$ (see Table 7). Their values, normalized by $\tau_{e}$, are shown in Figure 6, in respect to $Z_{S}$ (independently of the concentration, as all samples can be considered diluted), to assess the validity of the existing relationships presented in Section 1.

We first observe that within the experimental scatter, it seems that all the data follow the same curve, including the $\tau_{obs}$ parameter calculated from the zero-shear viscosity data. In particular, while the value of $\tau_{obs}$ does not depend on the molar mass of the long chains, which is in agreement with the well-established $M^{2}$ dependence of the CRR time, no significant difference appears between the different polymer chemistries. It should be noted, however, that we cannot exclude a slightly different behavior of the PI blends. Indeed, if we assume that $\tau_{obs}/\tau_{e}=K^{W}Z_{S}^{\alpha }$ with α = 3, as proposed in Ref. [3], we find that $K_{PS}^{W}=K_{PMMA}^{W}=K_{PBD}^{W}$=0.075, while $K_{PI}^{W}=0.050$, i.e., a factor 1.5 lower than for the other chemistries, in agreement with Ref. [3], in which a factor 2 was found between the CRR times of these PS and PI samples (the difference between the 1.5 and 2 factors is attributed to the influence of the materials parameters considered here to determine $\tau_{obs}$).

In Figure 6, it is also observed that the data could slightly differ from the scaling proposed, $\tau_{obs}\propto Z_{S}^{3}$ [3,14,22], showing that a lower dependence on $Z_{S}$ could better describe the whole range of data, as already noted in Ref. [3].

It must be noted that the curves shown in Figure 6 depend on the choice of the material parameters, which may affect their comparison. In order to avoid this source of uncertainty, we compare the behavior of the series of PS2810-Matrix, PI626-Matrix, PMMA234-Matrix, and PBD254-Matrix blends solely based on experimental data. The blends available in each of these series have the specificity to be composed of the same proportion of long chains diluted in matrices of various length. These long chains are all relaxing by CRR (see Figure 5). Therefore, as shown in Figure 7 for each series, it is possible to horizontally shift the storage modulus of the blends, $G_{blend}^{\prime }\left(\omega \right)$, by a factor λ, in order to overlap the terminal regime of a specific blend chosen as reference and containing a short chain matrix of mass $M_{S,ref}$. The terminal relaxation time $\tau_{d,blend}$ of the blend can thus be expressed as a function of the terminal relaxation time of the reference blend, $\tau_{d,blend_{ref}}$ and the shift factor, as $\tau_{d,blend}=\lambda \tau_{d,blend_{ref}}$. Therefore, if we consider that both samples are relaxing by a CRR process, the factor λ is equal to the ratio between $\tau_{CRR,L}$ and $\tau_{CRR,L_{ref}}$, the CRR time of the long chain in the reference blend. Or, equivalently, λ is equal to the ratio $\frac{\tau_{obs}}{\tau_{obs,ref}}$. Figure 7e shows the values of λ used to shift $G_{blend}^{\prime }\left(\omega \right)$ for each chemistry as a function of the ratio between $M_{S}$ and $M_{S,ref}$. One can observe that for all the series, the data follow the same trend, with λ scaling with ${\left(\frac{M_{S}}{M_{S,ref}}\right)}^{\alpha }$. Fitting with a linear regression, the value of α for each chemistry leads to $\alpha_{PS}\approx 2.32$, $\alpha_{PI}\approx 2.55$, $\alpha_{PBD}\approx 2.51$ and $\alpha_{PMMA}\approx 2.3$. Therefore, within experimental uncertainty, the value of lambda seems to be well described for all chemistries with α=2.5 (continuous black line) rather than with α=3 (dotted grey line). From this result, which is based only on experimental data, it is thus concluded that $\tau_{obs}\propto Z_{S}^{2.5}$.

If the CRR time is considered to be proportional to $Z_{S}^{2.5}Z_{L}^{2}$, one should find that a universal behavior of the long chain relaxation is recovered, whatever the sample chemistry might be. Since several blends contain a short chain matrix with the same number of entanglements (see Figure 4), we can further check this behavior by looking at their storage modulus normalized by $\frac{\upsilon_{L}\rho RT}{M_{L}}$ (see Equation (7)) as a function of $\omega \tau_{e}Z_{L}^{2}$. This way, the terminal relaxation time of the normalized curves is equal to $K^{W}Z_{S}^{\alpha }$ and, thus, only depends on $K^{W}$, since $Z_{S}$ is similar for all blends. The good overlap of the curves in the terminal regime (despite the small differences in the values of $Z_{S})$ shown in Figure 8 seems to confirm this universal behavior. Moreover, among these blends with the same $Z_{S}$, the blends PS2810-72.4 and PI626-17.6 also share the same $Z_{L}$. In such a case, the data do not have to be normalized by the number of entanglement segments of the long chain to be compared, and, according to the universal behavior of the samples, the terminal regime of the normalized storage moduli $\frac{G^{\prime }}{\upsilon_{L}G_{N}^{0}}$ as a function of $\omega \tau_{e}$ should superimpose. As shown in the Appendix A (see Appendix A), this is indeed observed. We therefore conclude that the constraint release Rouse time of long linear chains diluted in a short chain matrix seems to be fully described by the material parameters used in tube models, i.e., $G_{N}^{0}$, $M_{e}$ and $\tau_{e}$.

In Figure 6, the predictions of $\tau_{obs}$ obtained with Equations (4) and (5) are also presented [4,16]. While Equation (5) does not predict the correct evolution of $\tau_{obs}$ for matrices with a larger number of entanglements, the curve predicted by Equation (4) is close to the values of $\tau_{obs}$ obtained for the PI blends. However, Equation (4) underestimates the value of $\tau_{obs}$ for the other polymer chemistries. It seems therefore important to further investigate the relationship between $\tau_{obs}$ and the relaxation time of the matrix, $\tau_{d,S}$.

### 4.5. Relationship between τ_obs_ and Z_S_

In order to determine $\tau_{obs}$ as a function of $\tau_{d,S}$, the relaxation time of the short chain matrix should first be accurately determined. For a monodisperse linear polymer with Z entanglements, Likhtman and McLeish established from simulation data that the final relaxation time of the probe chain can be obtained from its reptation time by including the effect $f\left(Z\right)$ of contour length fluctuations, such that [20]:(21)$$\tau_{d}\left(Z\right)=\max\left(\tau_{e}Z^{2},3\tau_{e}Z^{3}f\left(Z\right)\right)/2$$
with:(22)$$f\left(Z\right)=1-\frac{3.38}{\sqrt{Z}}+\frac{4.17}{Z}-\frac{1.55}{Z^{1.5}}$$

On the other hand, the relaxation time of the matrices can be experimentally determined by $<\tau_{d}>=\underset{\omega \to 0}{\lim}\frac{G_{S}^{\prime }\left(\omega \right)}{\omega G_{S}^{\prime \prime }\left(\omega \right)}$ [3]. As shown in Figure 9, a very good agreement is found between these theoretical and experimental times for all the PS, PI, PMMA and PBD monodisperse samples. Furthermore, when the relaxation times are normalized by $\tau_{e}$ and the molar mass by $M_{e}$, all the data collapse into a master-curve, which further validates the values taken for these two material parameters.

As expected, using these experimental data to plot $\tau_{obs}$ in function of $\tau_{d,S}$ (see Figure 10a), it is observed that the data do not collapse onto a master-curve. However, if we consider that the release time of a short-long entanglement segment scales as:(23)$${\tau_{obs}\propto \tau }_{d,S}/Z_{S}$$
all the data superimpose into the same line of slope 1 (see Figure 10b).

This result further confirms that $\tau_{obs}\propto Z_{S}^{2.5}$. Indeed, within the range of molar masses investigated, the relaxation time of the entangled short matrices is well approximated by (see Figure 9):(24)$$\tau_{d,S}\sim 0.14\tau_{e}Z_{S}^{3.5}$$

Combining Equations (23) and (24), we therefore obtain:(25)$$\tau_{obs}\sim K0.14\tau_{e}Z_{S}^{2.5}$$
where K is a proportionality constant, which seems independent of the polymer chemistry according to the good superposition of all sets of data on Figure 10b.

The relationship $\tau_{obs}\propto \frac{\tau_{d,S}}{Z_{S}}$ can be explained as follows: a short chain cannot diffuse freely, by a Rouse process, since it is entangled. However, if we would assume that the short chain could move freely in all directions, one can determine an equivalent diffusion coefficient, $D_{3D}$, such that the time $\tau_{d,S}=\frac{Z_{S}^{2}N_{e}b^{2}}{D_{1D}}$, taken by the chain to diffuse along the tube axis and fully relax, is equivalent to the time it would take if we assume that the chain diffuses freely over a distance equal to its end-to-end distance, i.e., $\tau_{d,S}=\frac{Nb^{2}}{D_{3D}},$ or equivalently, $D_{3D}=\frac{Nb^{2}}{\tau_{d,S}}$. Considering that the constraint release time of an entanglement segment corresponds to the time the chain takes to freely explore the blob of an entanglement segment, we find: $\tau_{obs}=\frac{N_{e}b^{2}}{D_{3D}}=\frac{\tau_{d,S}}{Z_{S}}$.

Finally, in order to ensure that this release time is never shorter than the intrinsic Rouse time of an entanglement segment, $\tau_{e}$, the release time of a short-long entanglement segment is defined as:(26)$$\tau_{obs}=\tau_{e}+\frac{K\tau_{d,S}}{\max\left(2,Z_{S}\right)}$$

Based on the experimental data, the constant K, which is related to the efficiency of the constraint release process, was fixed to 1.4, which well agrees with the results obtained based on the slip-spring model^4^. In Equation (26), the condition $Z_{S}\geq 2$ accounts for the limiting case in which the polymer matrix is not entangled. Figure 11 shows the comparison between this equation (continuous black line) and the experimental data.

### 4.6. Critical Value of the Struglinski–Graessley Criterion for Dilute Binary Blends

In this Section, the critical value of the Struglinski–Graessley criterion, which determines the limit between relaxation via full CRR-like motion and relaxation by reptation, is discussed, based on new experimental data as well as on data available in the literature. To this end, we first recast all proposed critical values of $r_{SG}$ in the frame of the new criterion $r_{SG}^{*}$ proposed by Read et al. (see Equation (3)), in order to account for the influence of contour length fluctuations on the short matrix reptation [4].Consequently, the criterion proposed by Park and Larson, [19] $r_{SG}=\frac{Z_{L}}{Z_{S}^{3}}>0.1$, can be re-written as:(27)$$r_{SG}^{*}=\frac{Z_{L}}{Z_{S}^{3}f\left(Z_{S}\right)}>\frac{0.1}{f\left(Z_{S}\right)}$$
while according to the criterion proposed by Read et al. [4]:(28)$$r_{SG}^{*}>0.0254$$

These two criteria are compared to the experimental data in Figure 12. The different symbols are used to differentiate the blends for which the long chains were found to fully relax by CRR (+ symbols) (for which normalized $G_{L}^{\prime }\left(\omega \right)$ follow a Rouse-like relaxation on a wide frequency range, or as stated in the literature [5,14]) or not (o symbols).

It is seen that the criterion of Park and Larson results in a good description of the data, in spite of disregarding contour length fluctuations. Similar results are expected, based on the criterion proposed by Watanabe and co-workers [3,6,14], as it is based on the same scaling, the only difference being the presence of a pre-factor in the equation to account for the different chemistries. From Figure 12, one cannot conclude, however, that the CRR limit depends on the polymer nature. On the other hand, the limiting value proposed by Read et al. [4] underestimates the limit between reptation and CRR-like motion. However, it is interesting to note that if $\tau_{obs}$ is used instead of $\tau_{d,S}$ to describe the CRR time in the criterion, i.e.:(29)$$\frac{\tau_{d,L}}{\tau_{obs}Z_{L}^{2}}>1$$
the combination of this condition with the definition of $\tau_{obs}$ proposed by the authors (see Equation (4)) leads to a new critical value (with $\tau_{d,S}$ approximated by Equation (24)):(30)$$r_{SG}^{*}=\frac{Z_{L}}{Z_{S}^{3}f\left(Z_{S}\right)}>\frac{\tau_{obs}}{\tau_{d,S}}\approx 0.047*\left(1+\frac{1}{0.36}\sqrt{\frac{1}{0.047*0.14Z_{S}^{3.5}}}\right)$$
which is in good agreement with experimental data (see the black dashed line in Figure 12).

Similarly, in the present work, we propose a new critical value for $r_{SG}^{*}$ based on the condition (29) and the waiting time for a local CR-jump previously defined, $\tau_{obs}\approx \frac{1.4\tau_{d,S}}{Z_{S}}$ (see Equation (26)):(31)$$r_{SG}^{*}=\frac{Z_{L}}{Z_{S}^{3}f\left(Z_{S}\right)}>\frac{1.4}{Z_{S}}$$

This expression, which is based on $\tau_{obs}\propto Z_{S}^{2.5}$, leads to equally good results as Equation (30), based on $\tau_{obs}\propto Z_{S}^{3}$. Thus, these results suggest that if CLF are taken into account in the Struglinsky–Graessley criterion, its critical value is well defined by $\frac{\tau_{obs}}{\tau_{d,S}}$.

## 5. Modeling the LVE of Self-Unentangled Long Chains Diluted in a Short Chain Matrix

In this Section, the linear viscoelastic properties (LVE) of the different bidisperse blends presented in Section 3 are modeled, based on Equations (6)–(12), (21) and (26), to determine $\tau_{obs}$. The material parameters are given in Table 7 for the PS, PI, PMMA, and PBD polymers. Comparisons between predicted and experimental data are presented in Figure 13, Figure 14, Figure 15 and Figure 16 for the four types of chemistries.

A very good agreement is obtained for most of the samples, over the whole range of frequencies. This further validates the expression proposed to determine $\tau_{obs}$ and suggests that the CRR process can correctly be described based only on the material parameters. This last result should be further validated in the future, also based on other polymer architectures.

## 6. Conclusions

To conclude, an extensive dataset of PS, PI, PMMA, and PBD dilute bidisperse blends has been considered in order to examine the value of the release time associated with a short-long entanglement, $\tau_{obs}$, which governs the constraint release mechanism of the long chains. The value of $\tau_{obs}$ was first determined from experimental linear viscoelastic data following the method described in Ref. [3]. This allowed us to test and discuss the different scaling of $\tau_{obs}$ with the matrix molecular weight from the literature and to propose a new and simple expression, according to which $\tau_{obs}\propto Z_{S}^{2.5}$. Interestingly, it was found that, based on this expression, all the data of the CRR times collapse into a single curve within the experimental scatter, and the universal behavior of the long chain dynamics seems to be recovered for all polymer chemistries investigated in this work. Then, we tested the Struglinski–Graessley criterion. Instead of the original criterion $r_{SG}$, we considered the modified criterion $r_{SG}^{*}$ proposed by Read et al. to account for CLF of the matrix. It was shown that the critical value for $r_{SG}^{*}$ to obtain full CRR relaxation of the long chains is well described by the ratio $\frac{\tau_{obs}}{\tau_{d,S}}$, considering both $\tau_{obs}\propto Z_{S}^{2.5}$ and $\tau_{obs}\propto Z_{S}^{3}$. Finally, the new expression of $\tau_{obs}$ was implemented in a CRR model and tested on the different binary blends containing self-unentangled long chains. A very good agreement between experimental and predicted linear viscoelastic data was obtained for all polymer chemistries, supporting the new equation for $\tau_{obs}$ proposed in this study.

To conclude, we proposed a new simple expression for $\tau_{obs}$ that can be understood from a theoretical point of view and that can easily be implemented in tube models for different polymer chemistries. This is a first step towards the understanding of constraint release mechanisms in entangled bidisperse blends with self-entangled long chains.

## Figures and Tables

**Figure 1 polymers-15-01569-f001:**
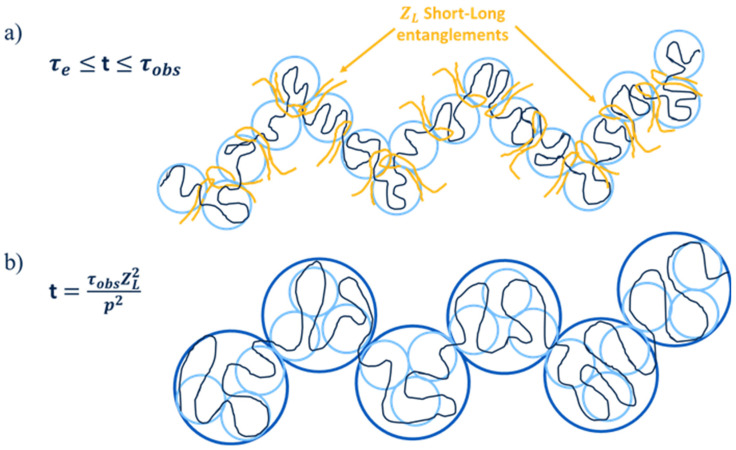
(**a**) Rouse relaxation modes of a self-unentangled long probe chain ($\upsilon_{L}Z_{L}$ < 2) diluted in a short chain matrix. The relaxation starts from the mode p=N and is followed by slower modes, down to the mode *p* = $Z_{L}$ corresponding to the entanglement segments (blue blobs). The latter are relaxed at time *t* = $\tau_{e}$, after which the intrinsic Rouse relaxation is stopped. (**b**) CRR relaxation process taking place for all the modes $1\leq p<Z_{L}$ (the blue blobs represent one of these modes). This process starts at time *t* = $\tau_{obs}$ and ends at time *t* = $\tau_{obs}Z_{L}^{2}$, which corresponds to the relaxation of the whole chain.

**Figure 2 polymers-15-01569-f002:**
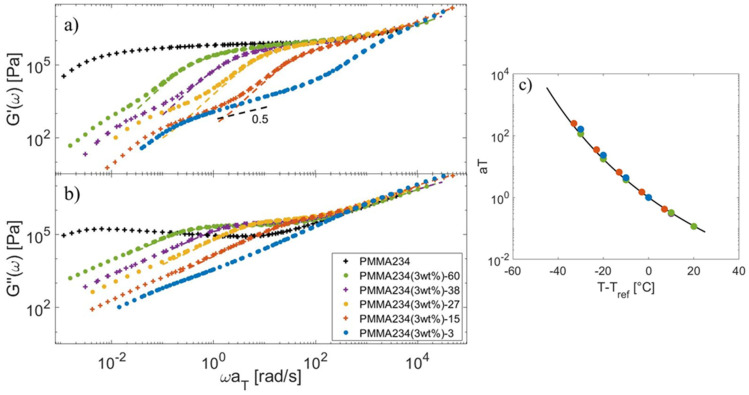
Storage (**a**) and loss (**b**) moduli under iso-$T_{g}$ condition of the PMMA234-Matrix blends (symbols) and of the short component in the monodisperse state (dashed lines), and (**c**) the corresponding shift factors at $T_{ref}=T_{g}+60\;{}^{\circ}C$, compared to the WLF Equation (black curve).

**Figure 3 polymers-15-01569-f003:**
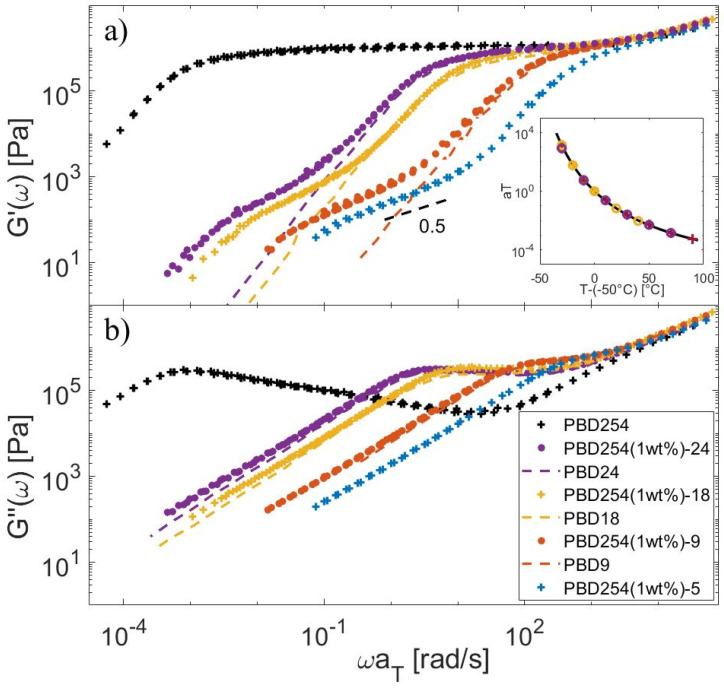
Storage (**a**) and loss (**b**) moduli shifted to $T_{ref}=-50\;{}^{\circ}C$, for the PBD254-Matrix blends (symbols) and for the short chain matrix in the monodisperse state (dashed line). Insert: corresponding shift factors.

**Figure 4 polymers-15-01569-f004:**
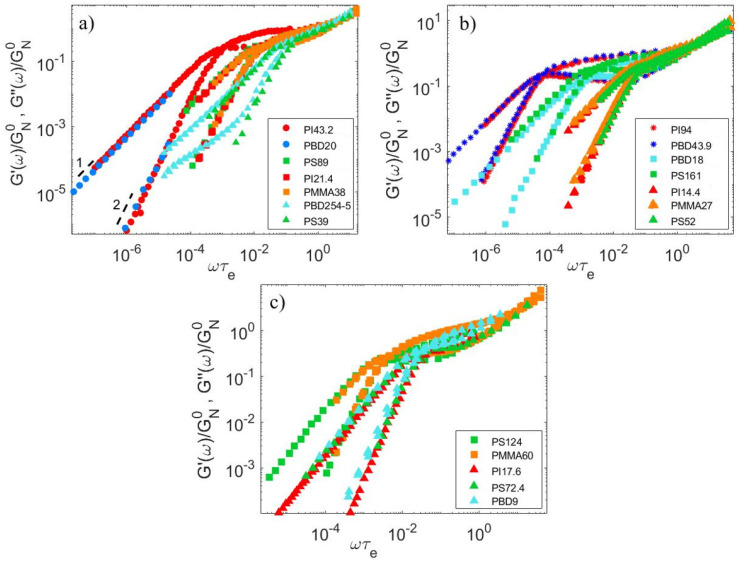
Comparison between normalized storage and loss moduli data of monodisperse samples with different polymer chemistries but the same number of entanglements Z, with (**a**) Z = 3 (▲), 6 (■) and 12 (●), with (**b**) Z = 4 (▲), 11 (■), and 26 (●), and with (**c**) Z = 5 (▲) and 9 (■) [6,13,14,16,17,18,20,21].

**Figure 5 polymers-15-01569-f005:**
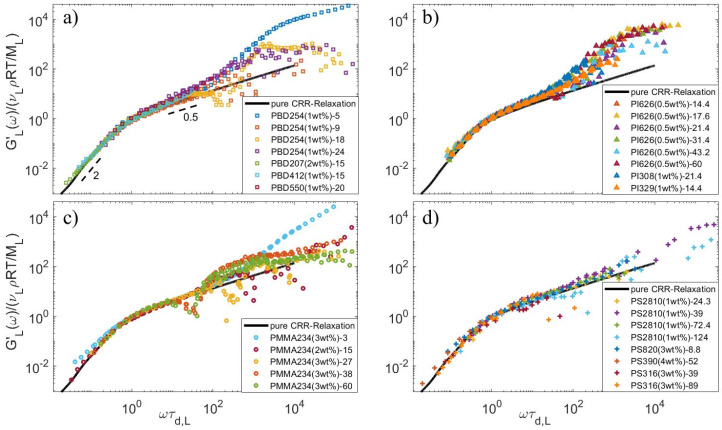
Normalized long chain storage modulus data of all available datasets under iso-$T_{g}$ conditions for (**a**) PBD, (**b**) PI, (**c**) PMMA, and (**d**) PS blends, compared to the expected storage modulus for a long polymer relaxing by a Rouse-like relaxation (black curve).

**Figure 6 polymers-15-01569-f006:**
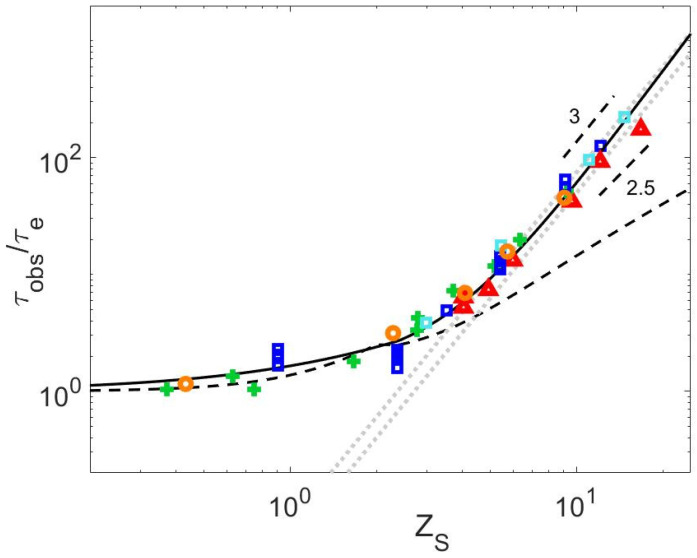
Normalized $\tau_{obs}$ data versus $Z_{S}$ for all available datasets for PS (green +), PI (red Δ), PMMA (orange ○), and PBD (blue ▯), compared to Equations (4) (black continuous line) and (5) (black dashed line) and the scaling $\tau_{obs}/\tau_{e}=K^{W}Z_{S}^{3}$ (grey dotted lines).

**Figure 7 polymers-15-01569-f007:**
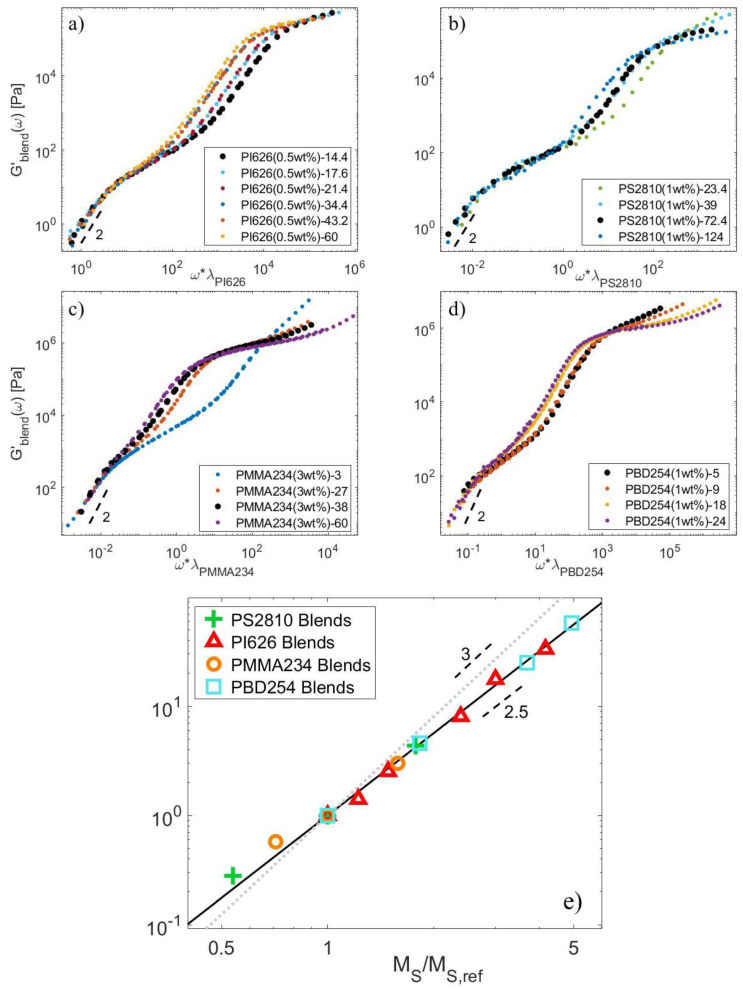
Storage moduli of (**a**) PI626-Matrix blends shifted by a factor $\lambda_{PI626}$ to fit on the terminal regime of PI626 (0.5 wt%)-14.4, (**b**) PS2810-Matrix blends shifted by a factor $\lambda_{PS2810}$ to fit on the terminal regime of PS2810 (1 wt%)-72.4, (**c**) PMMA234-Matrix blends shifted by a factor $\lambda_{PMMA234}$ to fit on the terminal regime of PMMA234 (3 wt%)-38, (**d**) PBD254-Matrix blends shifted by a factor $\lambda_{PBD254}$ to fit on the terminal regime of PBD254 (1 wt%)-5, (**e**) Shift factors λ for each series of blends plotted against $M_{S}/M_{S,ref}$ (symbols). The continuous black line corresponds to ${\left(\frac{M_{S}}{M_{S,ref}}\right)}^{2.5}$, while the dotted grey line corresponds to ${\left(\frac{M_{S}}{M_{S,ref}}\right)}^{3}$. The size of the symbols represents the uncertainty range in the value of λ obtained manually.

**Figure 8 polymers-15-01569-f008:**
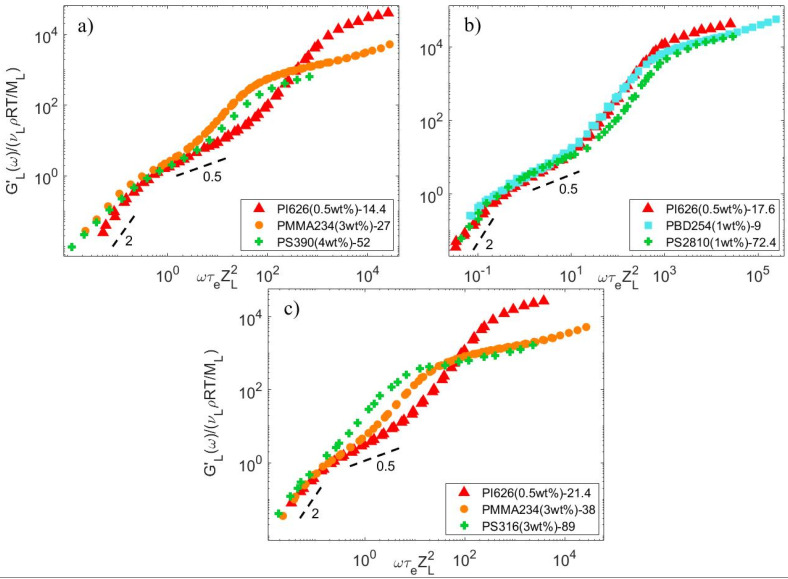
Normalized storage modulus of specific datasets with similar Z_S, such that (**a**) Z_S=4, (**b**) Z_S=5, and (**c**) Z_S=6.

**Figure 9 polymers-15-01569-f009:**
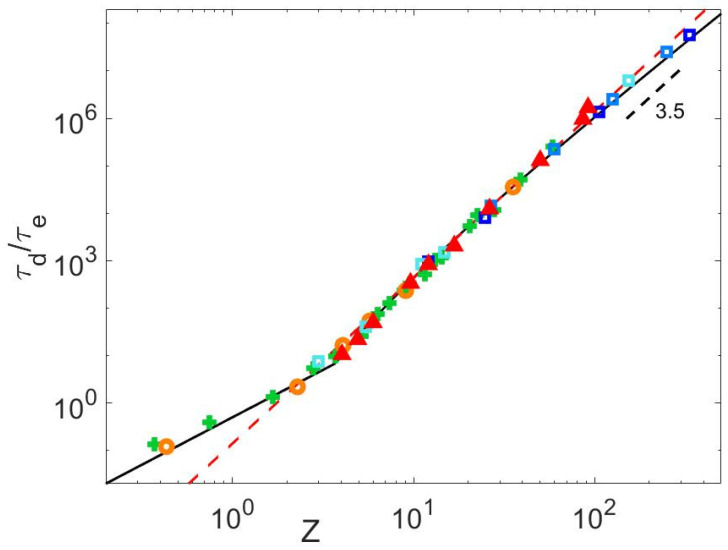
Normalized final relaxation times of monodisperse PS (green +), PI (red Δ) PMMA (orange ○) and PBD (blue ▯) samples from Refs. [14,24,25,26,27,28,30,40,41] and measured in this article, obtained from $<\tau_{d}>=\underset{\omega \to 0}{\lim}\frac{G^{\prime }\left(\omega \right)}{\omega G^{\prime \prime }\left(\omega \right)}$ [3], as a function of their number of entanglements, compared to the predictions of Equation (21) (black curve) and to the approximation $\tau_{d}/\tau_{e}\sim 0.14Z^{3.5}$ (dashed red curve).

**Figure 10 polymers-15-01569-f010:**
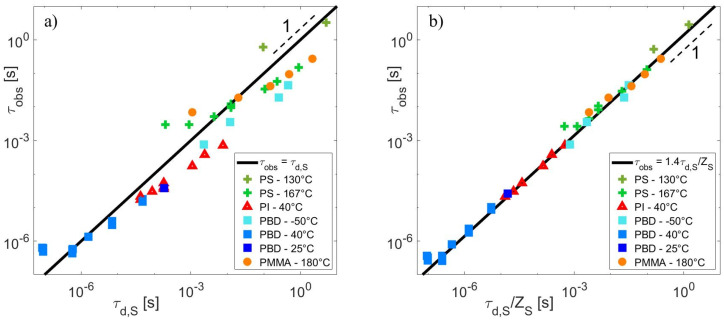
$\tau_{obs}$ versus $\tau_{d,S}$ (**a**) or $\tau_{d,S}/Z_{S}$ (**b**) for several self-unentangled blends of different polymer chemistries, from Refs. [14,24,25,26,27,28,30,40,41] and measured in this article.

**Figure 11 polymers-15-01569-f011:**
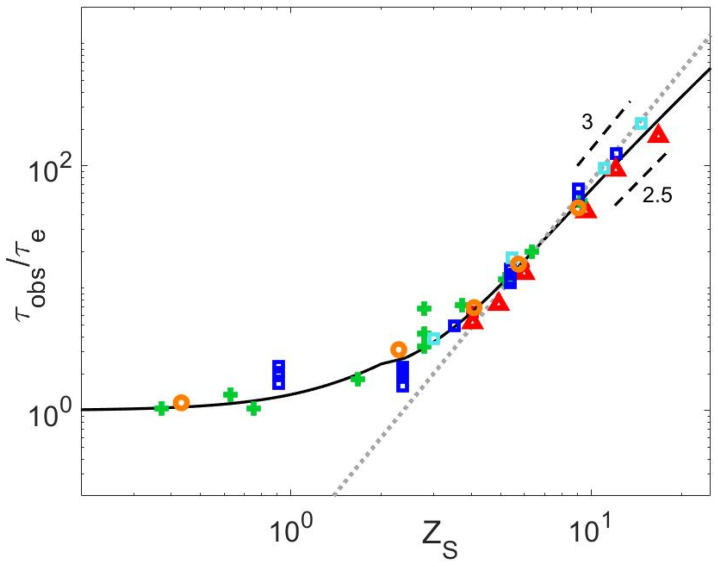
Normalized $\tau_{obs}$ data of all available datasets for PS (green +), PI (red Δ) PMMA (orange ○) and PBD (blue ▯) as a function of $Z_{S}$, compared to Equation (26) (continuous black line), and the scaling $\tau_{obs}=K^{W}\tau_{e}Z_{S}^{3}$ with $K^{W}=0.075$ (grey dotted line).

**Figure 12 polymers-15-01569-f012:**
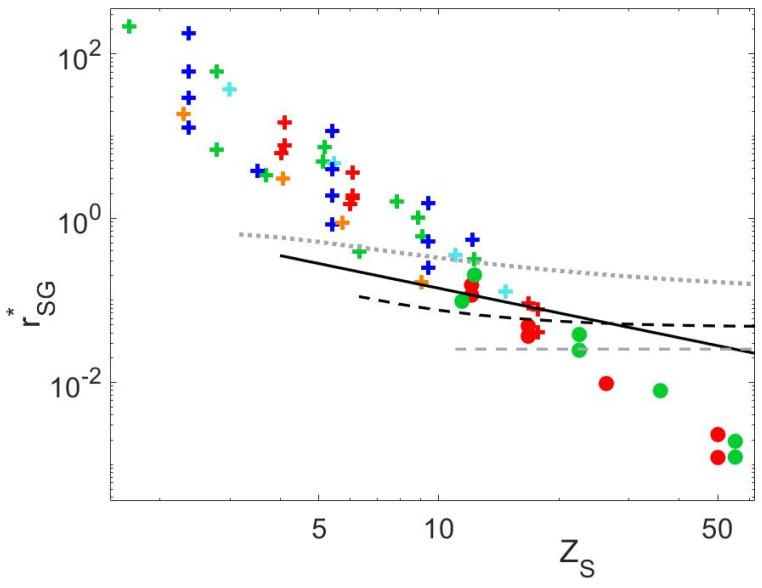
$r_{SG}^{*}$ for all PS (green), PI (red), PMMA (orange) and PBD (blue) blends available in this study and in the literature. The + and ○ symbols correspond to the data for which full CRR-like motion are observed or not, respectively. The data are compared to the critical value of $r_{SG}^{*}$ proposed by Park and Larson [19] (Equation (27) –dotted grey line), and by Read et al. [4] (dashed grey line). Furthermore, critical values based on Equation (30) (dashed black line), or based on Equation (31) (continuous black line) are given.

**Figure 13 polymers-15-01569-f013:**
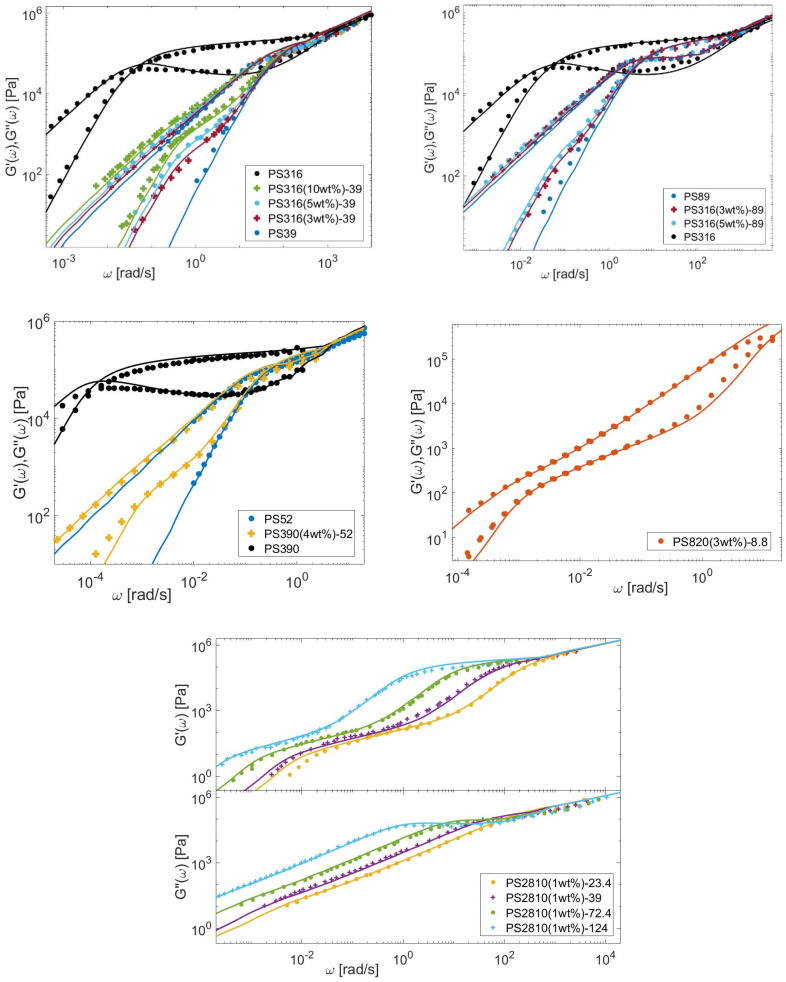
Comparison between predicted (continuous lines) and experimental (symbols) storage and loss moduli of the long PS chains in monodisperse state or diluted in various matrices (see Table 3).

**Figure 14 polymers-15-01569-f014:**
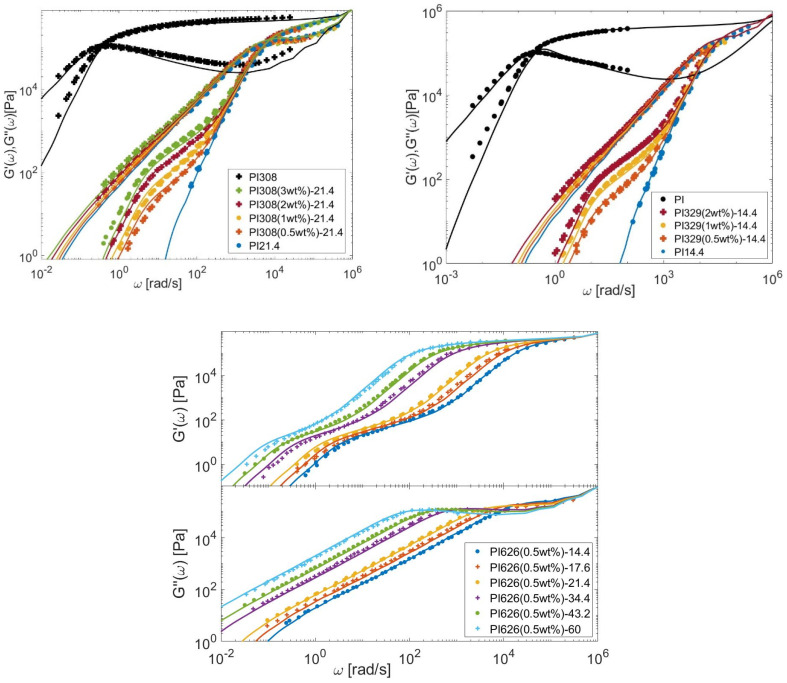
Comparison between predicted (continuous lines) and experimental (symbols) storage and loss moduli of the long PI chains in monodisperse state or diluted in various matrices (see Table 4).

**Figure 15 polymers-15-01569-f015:**
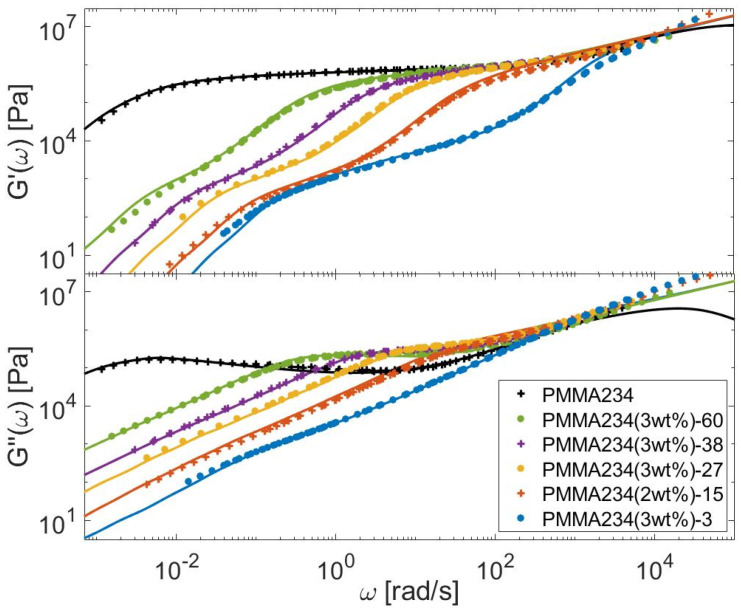
Comparison between predicted (continuous lines) and experimental (symbols) storage and loss moduli of the long PMMA chains in monodisperse state or diluted in various matrices (see Table 1).

**Figure 16 polymers-15-01569-f016:**
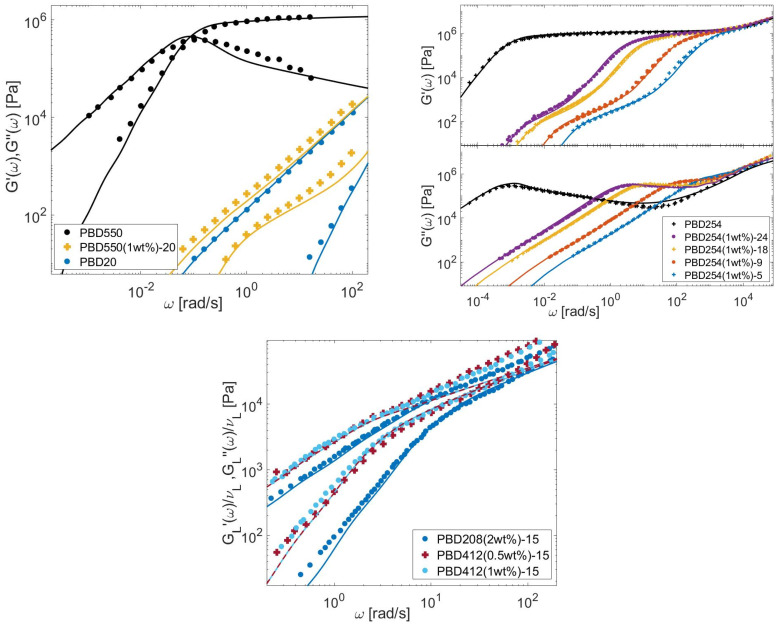
Comparison between predicted (continuous lines) and experimental (symbols) storage and loss moduli of the long PBD chains in monodisperse state or diluted in various matrices (see Table 2, Table 5 and Table 6).

**Table 1 polymers-15-01569-t001:** Molecular characteristics of purchased PMMA samples and bidisperse blends.

Sample	*M_L_* [kg/mol] (PDI)	*M_S_* [kg/mol] (PDI)	*υ_L_* [%]	*Z_L_υ_L_*	*T_g_*(*M_S_*) [°C]	*T_g_*(*blend*) [°C]	*T_ref_* [°C]
**PMMA234-60**	234 (1.11)	59.8 (1.08)	3	1.1	120	120	180
**PMMA234-38**		37.8 (1.11)	3	1.1	120	120	180
**PMMA234-27**		26.9 (1.09)	3	1.1	120	120	180
**PMMA234-15**		15.1 (1.08)	2	0.71	113	113	173
**PMMA234-3**		2.85 (1.09)	3	1.1	90	90	150

**Table 2 polymers-15-01569-t002:** Molecular characteristics of PBD blends.

Sample	*M_L_* [kg/mol] (PDI) 1,2:1,4 ratio	*M_S_* [kg/mol] (PDI) 1,2:1,4 ratio	*υ_L_* [%]	*Z_L_υ_L_*	*T_ref_* [°C]
**PBD254-24**	254 (1.02) 0.06:0.94	24.3 (1.01) 0.07:0.93	1	1.5	−50
**PBD254-18**		18.2 (1.01) 0.07:0.93	1	1.5	−50
**PBD254-9**		9.00 (1.02) 0.09:0.91	1	1.5	−50
**PBD254-5**		4.93 (1.03) 0.1:0.9	1	1.5	−50

**Table 3 polymers-15-01569-t003:** Bidisperse PS samples.

Sample	*M_L_* [kg/mol] (PDI)	*M_S_* [kg/mol] (PDI)	*υ_L_* [%]	*Z_L_υ_L_*	*T_ref_* [°C]	*T_g_*(*blend*) [°C]	Ref
**PS316-39**	316 (1.07)	38.9 (1.07)	3; 5; 10	0.68; 1.1; 2.3	167	103.9 *; 103.9 *; 104.1 *	[24]
**PS316-89**	316 (1.07)	88.6 (1.07)	3; 5	0.68; 1.1	167	106.6; 105.3 *	[24]
**PS2810-23.4**	2810 (1.09)	23.4 (1.07)	1	2.0	167	107; 100.9 *	[25]
**PS2810-39**	2810 (1.09)	38.9 (1.07)	1	2.0	167	107; 103.8 *	[25]
**PS2810-72.4**	2810 (1.09)	72.4 (1.06)	1	2.0	167	106.6; 105.1 *	[25]
**PS2810-124**	2810 (1.09)	124 (1.05)	1	2.0	167	106.6; 105.7 *	[25]
**PS820-9**	820 (1.02)	8.8 (1.1)	3	1.8	130	98.1	[26]
**PS390-52**	390 (1.06)	51.7 (1.03)	4	1.1	130	106.6; 104.5 *	[27]

* *T_g_* calculated from the Fox-Flory Equation.

**Table 4 polymers-15-01569-t004:** Bidisperse PI samples.

Sample	*M_L_* [kg/mol] (PDI)	*M_S_* [kg/mol] (PDI)	*υ_L_* [%]	*Z_L_υ_L_*	*T_ref_* [°C]	Ref
**PI308-21.4**	308 (1.08)	21.4 (1.04)	0.5; 1; 2; 3	0.43; 0.86; 1.7; 2.6	40	[28]
**PI329-14.4**	329 (1.06)	14.4 (1.03)	0.5; 1; 2	0.46; 0.92; 1.8	40	[30]
**PI626-14.4**	626 (1.06)	14.4 (1.03)	0.5	0.88	40	[14]
**PI626-17.6**	626 (1.06)	17.6 (1.04)	0.5	0.88	40	[14]
**PI626-21.4**	626 (1.06)	21.4 (1.04)	0.5	0.88	40	[14]
**PI626-34.4**	626 (1.06)	34.4 (1.04)	0.5	0.88	40	[14]
**PI626-43.2**	626 (1.06)	43.2 (1.03)	0.5	0.88	40	[14]
**PI626-60**	626 (1.06)	59.9 (1.05)	0.5	0.88	40	[14]

**Table 5 polymers-15-01569-t005:** Bidisperse PBD samples.

Sample	*M_L_* [kg/mol] (PDI) 1,2:1,4 ratio	*M_S_* [kg/mol] (PDI) 1,2:1,4 ratio	*υ_L_* [%]	*Z_L_υ_L_*	*T_ref_* [°C]	Ref
**PBD550-20**	550	20	1	3.3	25	[40]
**PBD208-15**	208 (1.01) 0.08:0.92	15.5 (1.10)	2	2.5	40	[41]
**PBD412-15**	412 (1.01) 0.08:0.92	15.5 (1.10)	0.5; 1	1.2; 2.5	40	[41]

**Table 6 polymers-15-01569-t006:** Bidisperse PBD samples from Ref. [41].

	*M_L_* [kg/mol] (PDI)	43.9 (1.01) −100°C	99.1 (1.01) −100 °C	208 (1.01) −100 °C	412 (1.01) −100 °C
*M_S_* [kg/mol] (PDI)					
**1.5 (/),** *T_g_*: −89 °C		3%; 5%	1%; 2%; 3%	0.5%; 0.8%; 1%	0.3%; 0.5%; 0.8%
**3.9 (1.10),** *T_g_*: −102 °C		2%; 3%	0.75%; 1%; 2%	0.5%; 0.75%; 1%	0.5%; 0.75%
**5.8 (1.06),** *T_g_*: −102 °C		0.8%; 1%; 2%; 3%	/	/	/
**8.9 (1.04),** *T_g_*: −102 °C		2%; 3%; 5%	0.5%; 1%; 2%	0.5%; 1%	0.2%; 0.5%
**15.5 (1.10),** (/)		/	0.75%; 1%; 2%	0.5%; 0.75%; 1%	0.1%; 0.2%; 0.5%; 0.75%

**Table 7 polymers-15-01569-t007:** Material parameters for the data sets.

Sample Set	*T_ref_* [°C]	*M_e_* [kg/mol]	*τ_e_* [s]	$G_{N}^{0}$ [kPa]	Ref of the Data
PS–130	130	14.0	0.39	230	[26,27]
PS–167	167	14.0	0.0025	210	[24,25]
PI–40	40	3.575	2.7 × 10^−6^	440	[14,28,29,30]
PMMA–180	180	6.6	0.006	720	/
PBD–25	25	1.65	2.1 × 10^−7^	1250	[40]
PBD–40	40	1.65	1.6 × 10^−7^	1250	[41]
PBD–−50	−50	1.65	2.0 × 10^−4^	1250	/

## Data Availability

The data measured for this study are available on request from the corresponding author.

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
