# Peer review of "Constraint Release Rouse Mechanisms in Bidisperse Linear Polymers: Investigation of the Release Time of a Short-Long Entanglement"

_polymers, 2023, doi:10.3390/polym15061569_

Round 1
Reviewer 1 Report
Report on Polymers-226069 by C. Hannecart et al.
The reptation mechanism proposed by de Gennes and Edwards, where a polymer diffuses along a tube defined by the constraints imposed by its surroundings, successfully describes the relaxation of long polymers in concentrated solution or melts. The central idea is that the non-crossing constraint allows chains to slide by each other but not to pass through each other. In this picture one tagged chain in the system moves under the constraints imposed by the surrounding chains, which build ``topological'' obstacles called entanglements.
This paper considers bi-disperse linear polymer melts, where long chain entanglements involve short chains. The relaxation dynamics strongly depends on the relaxation time of shorter chain.
The authors summarize several views on the relaxation dynamics of bi-disperse and investigate the relaxation time associated with short-long entanglement based extensive experimental data sets.
The paper is clearly written. The overall conclusions on the relaxation time scales are rationalized.
I would like to recommend publication of the manuscript after minor revision.
1. Please remind the reader that physical interpretation of the constant K in eq. (26)
2. It would be nice to include the guide lines for eyes to indicate exponents of scaling laws in several figures (Fig.2,3,4, ... 9).
3. For better readability, the x-axis labels should be spaced away from the tick labels.
Author Response
We would like to thank reviewer 1 for his/her positive comments. In the revised version, the readability of the pictures has been improved by including guidelines for characteristic scaling we wanted to emphasize and the x-labels have been spaced away from the tick labels. We also clarified in the manuscript that the constant K in equation (26) is related to the efficiency of the CR-process, corresponding to a local ‘hop’ of the chain (and not only to the bare release of a short-long entanglement segment).
Reviewer 2 Report
In this work, the authors first discussed the different approaches proposed to determine the average CRR time, ????, and compare them to a large set of experimental viscoelastic data. Based on this large set of data, the authors found that with respect to the molar mass of the short chain matrix, ????, follows a power law with an exponent close to 2.5, rather than 3. While this slight change in the power law exponent does not strongly affects the values of the constraint release times, the results obtained suggest the universality of the CRR process. The authors finally propose a new description of ????, which is implemented in a tube-based model. The accurate description of the experimental data obtained from the authors provides a good starting point to extend this approach to self-entangled binary blends. This work is systematically conducted and the agreement between theoretical predictions and experimental data is impressive.
Aside from traditional rheological measurements to determine the Rouse and repetation time, single molecule experiments could serve as a good complementary method to determine this time for entangled polymer solutions for comparison with rheological measurements (Zhou et al., Phys. Rev. Lett. 120, 267801, DOI:10.1103/PhysRevLett.120.267801), which might be of interest to the authors.
Author Response
We would like to thank reviewer 2 for his/her input and suggestion to compare rheological measurements with single molecule experiments. The article referenced by the reviewer focuses on the relaxation after extensional (nonlinear) deformation of DNA chains of constant length, diluted to different concentrations. In the scope of our article, we rather studied the linear response of long linear polymer chains of constant length diluted to the same concentration (so that they are self-unentangled), but in various matrix environments. Therefore, we could not directly compare the results or these two works. However, we keep in mind the article proposed by reviewer 2 for a future contribution, about the visco-elastic response of linear polymer solutions to uni-axial extensional flow.
Reviewer 3 Report
This paper investigates the relaxation mechanism of bidisperse linear polymer blinds in which the motion of long chains is confined by short chains. Viscoelastic properties of multiple polymer systems have been either characterized by rheology or sourced from literatures. The datasets have been well analyzed and a new quantitative relation have been proposed. The author has accurately reported their methods, results, and conclusions. Minor format corrections are needed to improve the reading:
(1) Equation alignment needs to be adjusted.
(2) Some sentences right after an equation should not be indented because it is not the beginning of a new paragraph (like line 49, line 70, etc.).
(3) In Section III. Material and methods, PMMA bidisperse blends is listed first, and bidisperse PBD blends is listed second. In Section IV. Results and Discussion, the sequence is the opposite.
(4) The caption of Figure 8 has different font and size than the rest of captions.(5) Figure 13-16: The description for lines is either missing in the legend or in the caption.
(6) All the references have been numbered twice.
Author Response
We would like to thank reviewer 3 for his/her positive comments, following which we improved the overall shape of the article (with particular attention to text indentation, equations, figures description and size, and to the order of the sequence in which data are presented). However, we did not find references numbered twice. Some of them have very similar titles, however they are different.